European citizens’ stance on limiting energy use for climate change mitigation

http://orcid.org/0000-0002-5687-7114 Mata Fernando 1 fernandomata@ipvc.pt
http://orcid.org/0000-0001-5404-8163 Nunes Leonel J. R. 2 3 4
1 CISAS—Center for Research and Development in Agrifood Systems and Sustainability, Instituto Politécnico de Viana do Castelo , Viana do Castelo , Portugal
2 DEGEIT, Departamento de Economia, Gestão, Engenharia Industrial e Turismo, Universidade de Aveiro , Aveiro , Portugal
3 GOVCOPP, Unidade de Investigação em Governança, Competitividade e Políticas Públicas, Universidade de Aveiro , Aveiro , Portugal
4 PROMETHEUS, Unidade de Investigação em Materiais, Energia e Ambiente Para a Sustentabilidade, Instituto Politécnico de Viana do Castelo , Viana do Castelo , Portugal
Jones Roger
Electronic publication date: 2023 Aug 7
Publication date: 2023
Volume: 11
Electronic Location ID: e15835
Received 2023 May 19; Accepted 2023 Jul 11
Copyright: © 2023 Mata and Nunes
Copyright year: 2023
Copyright holder: Mata and Nunes
License: This is an open access article distributed under the terms of the Creative Commons Attribution License, which permits unrestricted use, distribution, reproduction and adaptation in any medium and for any purpose provided that it is properly attributed. For attribution, the original author(s), title, publication source (PeerJ) and either DOI or URL of the article must be cited.
License URL: https://creativecommons.org/licenses/by/4.0/

Keywords: Climate change, Energy use, Perception, Attitudes, European citizens, Mitigation

Funding: The Foundation for Science and Technology (FCT, Portugal) CISAS UIDB/05937/2020 and UIDP/05937/2020 The Foundation for Science and Technology (FCT, Portugal) provided financial support to the CISAS UIDB/05937/2020 and UIDP/05937/2020, including the contract of Fernando Mata. The funders had no role in study design, data collection and analysis, decision to publish, or preparation of the manuscript.

==============================
Citizens’ attitudes and beliefs towards climate change are decisive in the adoption of mitigating measures. Limiting the use of energy in the context of climate change can be one of the mitigation measures, and therefore, understanding the position of the citizens towards it is important. With this aim, we used data from the 10th European Social Survey to relate the European citizens’ beliefs and attitudes on limiting the use of energy to tackle climate change. We have used variables related to demography and individuals’ perception of society and its policies. Statistical models were successfully fitted to data. Individuals with higher levels of trust in scientists have a higher degree of satisfaction with the national economies, are more worried about climate change and are more capable of assuming self-responsibility in climate change mitigation. These individuals have higher probabilities of believing that climate change mitigation could be achieved by limiting the use of energy. The EU citizens are, however, very skeptical in relation to the probability of many other citizens adopting measures to limit the use of energy.

Introduction

The stance of European citizens on limiting the use of energy in climate change (CC) mitigation is a topic of growing importance (Marquart-Pyatt et al., 2019; Xue et al., 2022), with the European Commission implementing mitigation targets under the Paris Agreement and the EU Climate and Energy Policy Framework 2030 (Kulovesi & Oberthür, 2020). This perspective constitutes a significant indicator of the general societal commitment to the fight global warming (Kotcher et al., 2021). At a time when Europe is striving to achieve carbon neutrality by 2050, as set out in the Paris Agreement, individual willingness to limit energy consumption is a critical element in achieving this goal (Chen et al., 2022). Limiting energy use not only contributes to the reduction of greenhouse gas emissions but also promotes energy efficiency (Akram et al., 2020). Thus, the stance of European citizens on this issue can directly influence energy and climate policies. A conscious and committed population can lobby governments to implement tougher and more effective measures for reducing energy consumption and transitioning to renewable sources (Poortinga et al., 2019; Rechsteiner, 2021). Furthermore, citizens’ commitment to limiting energy use can lead to significant changes in consumers’ behavior (Saari et al., 2021). This can result in increased demand for energy-efficient products and services, thereby encouraging innovation and sustainability in industries (Mahmood et al., 2022). The adoption of greener behaviors by consumers can also contribute to the creation of a circular and sustainable economy (Neves & Marques, 2022).

Another relevant aspect is that individual and collective awareness of the importance of limiting energy use can contribute to environmental education (Alam, 2023). This awareness can be passed on from generation to generation, creating a society more aware and responsible towards the environment (Dwivedi et al., 2022). Such awareness is essential to ensure the long-term sustainability of our planet (Mata et al., 2023). On the other hand, the stance of European citizens can have a significant international impact. Europe, being one of the most industrialized and developed regions in the world, has a crucial role to play in leading the fight against CC (Von der Leyen, 2019). A clear and demonstrable commitment from European citizens could serve as an example and influence the stance of other countries and regions (Blockmans, Hillion & Vimont, 2021). Additionally, a proactive stance by citizens can facilitate the implementation of public policies related to energy (Gregersen et al., 2020). With public support and energy efficiency policies, a transition to clean energy sources can be implemented more effectively and without significant resistance. This support can, therefore, accelerate the process of energy transition. Accordingly to the Social Cognitive Theory, efficacy beliefs are the main psychological factors for individual engagement behavior (Choi & Hart, 2021).

The stance of European citizens on limiting the use of energy reflects the growing awareness and urgency towards the climate crisis (Hoffmann et al., 2022; Mata et al., 2023). This sense of urgency is crucial to maintain the momentum necessary to combat CC. Only with a collective and determined commitment will be possible to mitigate the effects of CC and ensure a sustainable future for all (Zhenmin & Espinosa, 2019). This commitment reflects not only on the preservation of our environment but also on the ability to influence policies, shape consumer behaviors, boost innovation, and guide sustainability practices on a global level. Therefore, cultivating and sustaining this commitment is a fundamental step to ensuring an environmentally sustainable future and protecting our planet for future generations (El-Haggar & Samaha, 2019).

On a study conducted by Nunes (2022) based on a survey conducted in Portugal analyzing the perception of the impacts of CC and its causes most respondents recognize the need to mitigate greenhouse gas emissions as a key measure to reduce the impacts of CC. While not directly expressed, this perspective suggests the need to reduce energy consumption, since energy production is responsible for most greenhouse gas emissions (Lamb et al., 2021).

Using the environmental module of the European Social Survey (ESS) round eight executed in 2016, Verschoor et al. (2020) researched the public attitudes and beliefs towards CC and energy preferences. The fossil fuel sources were contrasted with the renewable and nuclear sources to understand public preferences in terms of CC implications, affordability, and security. The authors established a relationship between people’s beliefs and energy source preferences, with individuals showing higher levels of concern with CC preferring renewable sources.

Using the same survey, Gregersen et al. (2021) studied the relationship between public worries about CC and their energy-saving behaviors, concluding that more worried individuals tend to adopt and support energy-saving behaviors and policies. Also, using the same survey, Marquart-Pyatt et al. (2019) concluded that the predictors of green energy policy preferences correlate with those regarding the individual intention of energy-efficient behaviors. Similar conclusions were also obtained by Bouman et al. (2020) in a similar study.

This article aims primarily to understand the sensitivity of the European citizen in relation to the use of energy in the context of CC mitigation. The understanding of CC mitigation requires research on climate action and therefore, the beliefs and worries of individuals have paramount importance in policy definition and implementation (Steg, 2018). This study uses two questions taken from the ESS round 10 (2022) dependent variables: “Imagine a large number of people limit energy use, how likely would it be to reduce climate change?” and “How likely is it that a large number of people limit energy use?”. These questions were formulated to assess the perception of European citizens about the effectiveness and feasibility of limiting energy use as a strategy to mitigate CC. Through this approach, the study seeks to provide valuable insights into the role of energy consumption from the perspective of European citizens regarding CC, as well as evaluate potential barriers and opportunities for implementing mitigation policy-making strategies based on the reduction of the use of energy.

Materials and Methods

Data source

The open-access data set used in this study is the European Social Survey (ESS) (ESS ERIC, 2022a). In accordance with Article 23.3 of its Statutes, ESS ERIC subscribes to the International Statistical Institute’s Declaration on Professional Ethics, and therefore, Informed consent was obtained from all subjects involved in the study. Data were collected in 2022, from the 25th of May to the 18th of September. The ESS is in its 10th edition and took place in 25 different countries in Europe. It was implemented using mainly presential interviews, but also web questionnaires or videoconferences due to the COVID-19 pandemic.

The ESS is conducted by high-quality professionals including a multinational sampling and weighing experts’ panel. The interviewees are chosen using randomized sampling techniques with stratification by country, gender, social class, and type of community to obtain a fair representation of the universe being studied. The interviewers are also professionals trained to collect information without inducing biased responses. All questionnaires were cross translated into the local languages. Full survey specifications can be consulted through the ESS website (ESS ERIC, 2022b).

The ESS includes questions related to the life of European citizens, including attitudes and social behavior, general health and well-being, social indicators and conditions, attitudes and political behavior and ideology, inequality and social exclusion, inclusivity of minorities and equality, values and religion, national and cultural identity, media, linguistics and language, and family life (ESS ERIC, 2022c).

The pool of interviewees includes individuals over 15 years old, addressed in private households, independently of legal status, spoken language, nationality or citizenship. Interviews were conducted in Austria, Bulgaria, Switzerland, Czechia, Germany, Estonia, Spain, Finland, France, Greece, Croatia, Hungary, Iceland, Italy, Lithuania, Montenegro, North Macedonia, Netherlands, Norway, Poland, Portugal, Serbia, Sweden, Slovenia, and Slovakia. In total, 18,060 individuals were interviewed.

Variables in this study

In the present study, the aim is to understand the sensitivity of the European citizen to the use of energy in relation to CC mitigation. We have therefore selected the following questions as dependent variables (DV): “Imagine a large number of people limit energy use, how likely to reduce climate change?”; “How likely, a large number of people limit energy use?”

The survey randomly divided the interviewees into three groups, approximately representing 1/3 of the sample each. In each of the groups, the individuals responded to these two questions on three different scales. Group 1 responded on a Likert scale ranging from 0—not at all likely, to 10—extremely likely. Group 2 responded on a categorical scale (1—not at all likely, 2—not very likely, 3— likely, 4—very likely). Group 3 responded on a Likert scale ranging from 0—not at all likely, to 6—extremely likely.

As independent variables (IV) we have selected the following variables of interest.

Demographic: ‘Gender’, ‘Years of full-time education completed’, and ‘Age’.

Perception of society and its policies: ‘Satisfaction with the state of the economy’, ‘Trust in the legal system’, ‘Satisfaction with the democratic system’, and ‘Satisfaction with the government’.

CC related: ‘How worried are you about climate change?’, ‘To what extent do you feel a personal responsibility to try to reduce climate change?’, and ‘Trust in scientists’.

The IVs demographic questions had an option of women or men, for ‘Gender’ and years was the answering unit for ‘Years in full-time education completed’ and ‘Age’. The other IVs were responded to on a scale from 0—not at all, up to 10—completely: ‘Satisfaction about the national economy’, ‘Satisfaction about the national government; ‘Satisfaction with the state of the democracy in the country’, ‘Trust in the legal system’, and ‘To what extent do you feel a personal responsibility to try to reduce climate change?’. The questions about ‘Trust in the legal system’ and ‘Trust in scientists’ were responded on a scale from 0—no trust to 10—completely trust. The question ‘How worried are you about climate change?’ was responded to on a Likert scale (1—not at all worried, 2—not very worried, 3—somewhat worried, 4—very worried, and 5—extremely worried).

All the questions surveyed had as answering options ‘Didn’t give an answer’, ‘Don’t know’, or ‘Refused to answer’. These cases did not enter the statistical analysis.

Statistical analysis

From the three groups of response scales for the DV previously identified, we have chosen the second group, since the four categories allow an easier graphic representation of the models, facilitating its interpretation.

To analyze the data we used a multinomial regression with a logit link. The selection of variables followed the backward stepwise procedure. The models were accessed via the −2 Log likelihood χ2 test and the parameters of the models via the Wald chi-square test. The odds ratios of the variables in the model were also calculated, together with their Wald confidence intervals. The level of significance was set to p < 0.05 in all the statistical tests. The procedures were carried out using the NOMREG routine of the software IBM Corp.® SPSS® Statistics, (Armonk, NY, USA. Version: 28.0.1.1 (15)).

For graph production, we used Microsoft® Excel® for Microsoft 365 MSO (version 2204 Build 16. 0. 15128. 20240) 64-bit.

Results

Descriptive statistics

Of the 18,060 interviewees, 5,925 were allocated to Group 2, the group of interest in this study.

Table 1 reports the descriptive statistics. The DVs were responded to on a categorical scale and the response densities for each category can be observed in Fig. 1.

Table 1 Descriptive statistics included in the models: answers to the dependent variables were given in a categorical scale and are represented in Fig. 1.

	Variables§	
	1	2	3	4	5	6	7	8	9	
N	Valid	5,907	3,931	5,877	5,864	5,863	5,891	5,991	5,890	5,925	
Omitted	84	2060	114	127	128	100	0	35	0	
Mean	6.79	4.84	6.78	4.67	4.28	4.88	5.70	50.97	14.45	
Median	7	5.00	7.00	5.00	4.00	5.00	6.00	52.00	12.00	
Standard error	2.460	2.870	2.450	2.466	2.710	2.672	2.827	18.611	10.330	
Minimum	0	0	0	0	0	0	0	15	0	
Quartile	25	3.00	5.00	3.00	2.00	3.00	4.00	3.00	36	11	
50	5.00	7.00	5.00	4.00	5.00	6.00	3.00	52	12	
75	10	7.00	9.00	7.00	6.00	7.00	8.00	66	16	
Maximum	10	10	10	10	10	10	9	90	40	
Notes:

§ Variables on a 0 to 10 scale.

Variables: 1—‘Trust in the legal system’, 2—‘Trust in scientists’, 3—‘Satisfaction with the state of the national economy’, 4—‘Satisfaction with the national government’, 5—‘Satisfaction with the democratic system’, 6—‘To what extent do you feel a personal responsibility to try to reduce climate change?’, 7—‘How worried are you about climate change? ’, 8—‘Age’, 9—‘Years of full-time education completed’; Refusal to answer, not giving an answer, or answering ‘don’t know’ were omitted from the analysis.

Figure 1 Distribution of answers to the questions used as dependent variables.

Model for the answers to the question: imagine large numbers of people limit energy use, how likely to reduce climate change?

The multinomial model was successfully fitted to the categorical DV. The respective parameterization is shown in Table 2. The model is also represented in Fig. 2. Due to the multidimensionality of the model, the different graphs in the figure vary for the identified variable and are fixed in their means for the other variables.

Table 2 Model parameterization for the question ‘Imagine large numbers of people limit energy use, how likely to reduce climate change?’ The model is significant, −2 Log likelihood χ2 = 814, 12 df, p < 0.001.

					95% Wald CI	
Response	Parameters	β	Std error	Exp (β)	Lower	Upper	
Not at all likely	Intercept	6.892***	0.040				
Trust in scientists	−0.082*	0.115	0.921	0.854	0.993	
State of economy	−0.110**	0.040	0.896	0.829	0.968	
Worried about CC	−1.373***	0.342	0.253	0.202	0.317	
Self-responsibility	−0.447***	0.027	0.640	0.591	0.693	
Not very likely	Intercept	6.609***	0.025				
Trust in scientists	−0.072**	0.080	0.931	0.882	0.982	
Worried about CC	−1.053***	0.313	0.349	0.298	0.408	
Self-responsibility	−0.280***	0.025	0.756	0.717	0.797	
Likely	Intercept	4.264***	0.023				
State of economy	0.051*	0.025	1.052	1.007	1.100	
Worried about CC	−0.593***	0.040	0.553	0.480	0.635	
Self-responsibility	−0.118***	0.115	0.888	0.846	0.933	
Notes:

The category ‘Very likely’ is used as the reference in the model; state of economy—‘Satisfaction with the state of the national economy’, worried about CC—‘How worried are you about climate change, self-responsibility—‘To what extent do you feel a personal responsibility to try to reduce climate change?’.

* p < 0.05.

** p < 0.01.

*** p < 0.001.

Figure 2 Model for the question: ‘Imagine large numbers of people limit energy use, how likely to reduce climate change?’

Each graph represents the variation of one of the main effects of the independent variable while fixing the others in their mean values.

Overall, almost half of the interviewees responded ‘likely’ to the question. Together with the ‘very likely’ answer comprise around 61% of the interviewees. There is, therefore, a 39% share of individuals that disbelieve in CC mitigation with the limitation of energy use. Out of these, only 6% is in the extreme position (‘not at all likely’).

People with a positive perception (responding ‘likely’ or ‘very likely’) agreeing that a limitation to the use of energy may reduce CC show an increasing level of trust in scientists. The opposite is observed for those more skeptical (responding ‘not very likely’).

Individuals with a higher degree of satisfaction with the national economy agree with the likelihood that limiting energy use will mitigate CC. However, for the individuals responding ‘very likely’, the inverse is observed, as well as for those responding ‘not very likely’.

The probability of an individual being worried about the CC increases if these individuals also believe that limiting the use of energy also impacts CC mitigation. The opposite is observed for those less worried.

Individuals feeling self-responsibility in trying to reduce CC, have higher probabilities of also believing that limiting energy use may mitigate CC. ‘Self-responsibility’ and worries about CC correlate positively (Spearman’s rho = 0.414, p < 0.001).

Model for the answers to the question: how likely, a large number of people limit energy use?

The multinomial model was successfully fit to the categorical DV. The respective parameterization is shown in Table 3. The model is also represented in Fig. 3. Due to the multidimensionality of the model, the different graphs in the figure vary for the identified variable and are fixed in their means for the other variables.

Table 3 Model parameterization for the question ‘How likely, a large number of people limit energy use?’ The model is significant, −2 Log likelihood χ2 = 351, 21 df, p < 0.001.

					95% Wald CI	
Response	Variable	β	Std. error	Exp (β)	Lower	Upper	
Not at all likely	Intercept	4.749***	0.660				
	State of economy	−0.114*	0.057	0.892	0.797	0.998	
	Worried CC	−0.547***	0.124	0.579	0.454	0.738	
	Self-responsibility	−0.248***	0.043	0.780	0.716	0.849	
	Age	−0.011*	0.006	0.989	0.978	1.000	
	Years in education	0.092**	0.030	1.097	1.034	1.163	
Not very likely	Intercept	3.815***	0.612				
	Worried CC	−0.409***	0.115	0.664	0.531	0.832	
	Self-responsibility	−0.136***	0.040	0.873	0.807	0.944	
	Years in education	0.112***	0.027	1.119	1.061	1.180	
Likely	Intercept	2.399***	0.625				
	Worried CC	−0.255**	0.117	0.775	0.616	0.975	
Notes:

The category ‘Very likely’ is used as the reference in the model; State of the economy—‘Satisfaction with the state of the national economy’, worried about CC—‘How worried are you about climate change, self-responsibility—‘To what extent do you feel a personal responsibility to try to reduce climate change?’, years in education ‘Years in full-time education completed’.

* p < 0.05.

** p < 0.01.

*** p < 0.001.

Figure 3 Model for the question: ‘How likely, a large number of people limit energy use?’

Each graph represents the variation of one of the main effects of the independent variable while fixing the others in their mean values.

Individuals more worried about CC are more optimistic in relation to believing a large number of people accept a limitation in energy use.

People willing to assume self-responsibility in CC mitigation are also more optimistic in relation to believing a large number of people accept a limitation in energy use.

More educated people are more pessimistic in believing that a large number of people will limit energy use for CC mitigation.

Age influences the extreme option ‘not very likely’, and the probability to choose this option while answering the question decreases with age. This, however, is accomplished mainly at the expense of the answer ‘not very likely’, therefore still on the pessimistic side. On the other hand, age has a slightly positive effect on the optimistic options (‘likely’ and ‘very likely’), with older people having a higher probability of choosing these options.

The satisfaction in relation to the national economy has a decreasing effect on the probability of an extremely pessimistic option (‘not at all likely’), at the cost of an increase of the option ‘not very likely, therefore still on the side of the individuals with pessimistic feelings in relation to a large number of people limiting energy use.

Discussion

The data analyzed in this study provides a comprehensive insight into the European citizens’ perceptions of CC, their trust in scientists, their satisfaction with the national economy, government, and democratic system, and their beliefs about limiting energy use as a mitigation strategy. In general, the data reveals a positive correlation between individuals’ trust in scientists, satisfaction with the national economy, and their belief in the effectiveness of limiting energy use for CC mitigation.

The perception of personal responsibility and worry about CC also show significant relationships with the belief in limiting energy use as an effective strategy. It implies that individuals who feel a strong personal responsibility to reduce CC, and who are more worried about it, are more likely to support energy limitation. On the other hand, those who are skeptical about the effectiveness of limiting energy use tend to show lower levels of trust in scientists, diminished satisfaction with the state of the national economy and tend also to be less worried about CC. These results highlight the importance of addressing these factors to enhance public engagement in CC mitigation efforts.

The analysis reveals a nuanced influence of education and age on people’s belief in the effectiveness of limiting energy use. More educated individuals and older individuals tend to be more pessimistic about the likelihood of large numbers of people accepting limitations in energy use, despite being more likely to accept such limitations themselves. This suggests that while education and age might increase individuals’ acceptance of energy limitations, they may also increase skepticism about others’ willingness to do the same. The results also show a relationship between satisfaction with the national economy and the belief in the effectiveness of limiting energy use. This underscores the role of economic factors in shaping public attitudes toward CC mitigation strategies. It suggests that improving the economic conditions and addressing economic insecurities might be important for fostering public support for energy limitation strategies. Lastly, the role of trust in legal systems and satisfaction with the national government, when associated with age, seems to influence the belief that people limit energy use. This suggests that fostering trust in national institutions and improving government performance might also be important for enhancing public support for CC mitigation strategies.

With Russia’s invasion of Ukraine, the EU faced new challenges to balance its energy mix. The dependency on Russia’s gas has proved to be politically wrong, and the EU countries had to phase out the Russian gas supply. Meanwhile, Russia retaliated and cut the supply to the EU countries. The sudden search for alternative sources of gas, but also oil, pushed up energy prices around the world (Osička & Černoch, 2022). The EU passed legislation on August 2022 where EU countries have agreed on a reduction of 15% in the consumption of gas (EU, 2022).

Energy consumption from fossil fuels is identified as a main driver in carbon emissions and therefore CC (Hickel et al., 2021). Limitations to energy use are, therefore, measures of CC mitigation. The first model in this study retrieves information about the EU citizens’ understanding of the limitations in energy use and its impact on CC mitigation.

The answers to the question “Imagine large numbers of people limit energy use, how likely to reduce climate change?” show that most of the EU citizens (61%) agree that limiting energy use, mitigates CC. To tackle CC effectively, however, we need compromised action at a global level from every citizen. As such, the figure of 61%, with 39% skeptical, is not very good news.

Individuals acknowledging that energy use reduction contributes to the mitigation of CC, show higher levels of trust in scientists. This result agrees with those obtained by other studies (e.g., Cologna & Siegrist, 2020; Hornsey et al., 2016).

The importance of trust in scientists to tackle CC has been identified and discussed. Lay audiences’ trust in scientists is essential to avoid disruptive information from doubtful and dismissive individuals (Goodwin & Dahlstrom, 2014). Positions taken by populist politicians have been identified as disruptive of trust in scientists in the USA the EU and many other countries (Hamilton, Hartter & Saito, 2015). Motivations are political, industrial, and obviously economic (Cordero, Centeno & Todd, 2020; Hegerl et al., 2019). The undermining of scientific evidence of CC has been led by the fossil fuel industry (Leal Filho et al., 2021). Trust in science and scientists has been identified as the foremost item in public acceptance of proposed actions to mitigate CC (Lucas, Leith & Davison, 2015), rather than education (Gauchat, 2012; Wynne, 1992). A study (Brulle, Carmichael & Jenkins, 2012) in the USA found that scientific information has a small but significant effect in shaping social concerns with CC. Scientific articles are generally not read by large audiences and do not influence the public but reports and articles in lay press or science magazines do. Scientists may be at a disadvantage, however, in relation to politicians, as these have wider audiences.

Individuals with a higher degree of satisfaction with their national economies responded more favorably to the limitation of energy in CC mitigation. Other studies (Britto, Dehler-Holland & Fichtner, 2022; Poortinga et al., 2003) have shown that investments in energy-efficient equipment are positively correlated with wealth. It has also been shown (Poortinga et al., 2003) that wealthy individuals are less attracted to directly reducing energy use. Poortinga et al. (2003) found that home technical improvements are more accepted by individuals with higher income, while those with lower income prefer behavioral, such as transport, measures. An explanation for this behavior is the fact that daily actions may reduce comfort, while the decision taken in a one-time purchase is seen as an investment that may even increase comfort in the long run (Udalov, 2019). Purchasing is a one-time effort, while behavioral changes require constant efforts (Gardner & Stern, 1996).

Economic conditions obviously impact attitudes, and it was shown (Alló & Loureiro, 2014) that willingness to pay for the control of CC is higher in developed countries. People from countries in development prefer policies with mechanisms that do not increase consumer prices.

In our study, individuals with a lower degree of satisfaction with the national economies are less likely to believe in energy use restrictions to mitigate CC. These unsatisfied individuals are more likely to feel economic constraints, and therefore, already limit energy use to the very essential. It should however be noted that wealth has not been identified as a factor of belief and willingness to act for CC mitigation (Alló & Loureiro, 2014). Poorer individuals tend to point out the investment in clean energy production infrastructure as a solution for CC mitigation (Alló & Loureiro, 2014; Mayer & Smith, 2019; Poortinga et al., 2003).

Studies have also shown that individuals with a higher perception of the CC risks, and more worried about the CC consequences, incite change in behaviors and increase their willingness to pay for CC mitigation (Brulle, Carmichael & Jenkins, 2012; Mata et al., 2023; Mayer & Smith, 2019). Other studies (Leiserowitz, 2006; O’Connor, Bard & Fisher, 1999; Smith & Leiserowitz, 2014) have shown the relationship between worried individuals about CC and the belief in the adoption of CC mitigating measures. Smith & Leiserowitz (2014) found that “worried” is one of the main emotions felt by Americans when thinking about global warming, and therefore the CC problem. While applying their data in a model to predict support for national climate and energy policies, these authors found that “worry” is the single strongest predictor. Worry motivates cognitive and analytical processing of risk perception, more intensely. Mata et al. (2023) found the same in relation to European citizens.

Self-responsibility feelings in trying to reduce CC are positively correlated with worries about CC. In that sense and as expected, more self-responsible individuals are more likely to adopt measures such as limiting energy use in CC mitigation. Smith & Leiserowitz (2014) have also found that positive feelings, namely hope and interest, impact positively the adoption of CC mitigation measures. The sense of doing the “right thing” has previously been associated with the adoption of CC mitigation measures (Markowitz & Shariff, 2012; Roeser, 2012). Several other authors have also found that feelings of self-responsibility are predictors of positive attitudes toward CC mitigation (Mata et al., 2023; McCright, Dunlap & Marquart-Pyatt, 2016a, 2016b; van der Linden, 2015).

The set of data analyzed is robust as it contains a large sample size from different countries, and has also a well-diversified spectrum of citizens, however, limitations may be identified. There are multiple other factors that can also possibly affect and interact with those considered in the present study. For example, while studying the effort to save energy as a CC mitigation tool, Boto-García & Bucciol (2020) found that income and religious belief are positively correlated with responsibility but negatively correlated with CC mitigation behavior. An important aspect of the 10th ESS survey is that it was carried out during the COVID-19 pandemic in 2022, which somehow could impact the results as people under the circumstances individuals may be more sensible about global issues and the effects of human activity. This could potentially modulate the degree of concern about CC and the perception of its threat. Another aspect was the lack of consideration given to the answers ‘Don’t know’, ‘Didn’t give an answer’, and ‘Refused to answer’. These answers did not enter the models. The number of entries in the dataset is small, therefore an outcome of these results was not considered.

Conclusions

This study aimed to provide insights into the role of energy consumption from the perspective of European citizens in relation to CC. The data reveal a positive association between individuals’ trust in scientists, satisfaction with the national economy, and their belief in the effectiveness of limiting the use of energy in CC mitigation. Individuals with higher levels of trust in scientists, a higher degree of satisfaction with the national economies, more worried about CC, and more capable of assuming self-responsibility in CC mitigation, have higher probabilities of believing that CC mitigation could be achieved by limiting the use of energy. These individuals are, however, very skeptical in relation to the probability of a large number of other citizens adopting mitigation measures to limit the use of energy.

Additional Information and Declarations

Competing Interests

Author Contributions

Data Availability

The authors declare that they have no competing interests.

Fernando Mata analyzed the data, prepared figures and/or tables, authored or reviewed drafts of the article, and approved the final draft.

Leonel J. R. Nunes analyzed the data, prepared figures and/or tables, authored or reviewed drafts of the article, and approved the final draft.

The following information was supplied regarding data availability:

The data is available at the European Social Survey data portal: https://ess-search.nsd.no/en/study/172ac431-2a06-41df-9dab-c1fd8f3877e7.

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
