# Peer review of "European citizens’ stance on limiting energy use for climate change mitigation"

_PeerJ, doi:10.7717/peerj.15835_

## Round 0.1 · original submission · Minor Revisions

Apologies for the time in review - people are very busy at the moment. I echo the comments made by the reviewers.

Reviewer 1 ·

Basic reporting

This paper addresses a topic which is of interest to PeerJ readers as it explores by survey data the attitudes of European citizens on energy use and CC mitigation. Due to the new ESS10 data utilised, the originality of the paper seems good. The paper also displays awareness of the issue area and refers to previous empirical studies.

Even though the paper seems interesting, its originality and contribution could be strengthened by a stronger framing of the issue. The literature in the background of the paper is basically somewhat thin. The introduction also ignores the contexts of knowledge. Thus:
1.I suggest that the author extends the framing with journal articles based on the ESS Round 8, where the same questions were also used (instead, they were not included in the Round 9). There are indeed a lot of excellent papers supporting the justification of this paper.
2. I do understand the choice to focus on European attitudes in general, but I also consider it a bit problematic. The least the author should do is to be punctual with the contexts. For instance, the results presented on lines 83-102 are based on a data from Portugal only but that is not told to a reader. Thus, I advise to consider whether that many lines are needed for the results from that paper and also to absolutely explicate the context of the results.

Experimental design

The paper utilises ESS10 survey data consisting of 5925 respondents. The data is accurate though the socio-demographics are under-utilised (only age and education) and the results are presented by two tables and two figures. There are some unclear sentences in the results section -- sentences in lines 189-90, 195-6, 212-2, and 221 should be clarified.

Validity of the findings

Overall, the main ambition of the paper is basically empirical, i.e., there is no strong theoretical contribution to be found. Some practical implications are discussed.

Additional comments

Regarding the language, the proof-reading is required.

Reviewer 2 ·

Basic reporting

The manuscript is clearly stated, and methods are well explained. Here are some of my comments for the introduction. As the study was based on survey, there will be potential bias in the answers to question contain sensitive behaviors. There are different sampling techniques that could resolve these issues (randomized response technique), which could be mentioned in the introduction.

Warner SL. Randomized response: A survey technique for eliminating evasive answer bias. Journal of
the American Statistical Association. 1965; 60(309):63–69

Greenberg BG, Kuebler RR Jr, Abernathy JR, Horvitz DG. Application of the randomized response
technique in obtaining quantitative data. Journal of the American Statistical Association. 1971; 66
(334):243–250

Lensvelt-Mulders GJ, Hox JJ, Van der Heijden PG, Maas CJ. Meta-analysis of randomized response
research: Thirty-five years of validation. Sociological Methods & Research. 2005; 33(3):319–348

Cao, M., Breidt, F.J., Solomon, J.N., Conteh, A. and Gavin, M.C., 2018. Understanding the drivers of sensitive behavior using Poisson regression from quantitative randomized response technique data. PloS one, 13(9), p.e0204433.

Experimental design

As mentioned above for question, for sensitive questions, the interviewee would potentially not like to answer, which could result a "don't know" or "did not give an answer". These answer may not me similar distributed as all the other answers which mean the missingness may be informative, this should be discussed.

The multinomial regression assumption should be justified as the quantitative answers could potentially be modelled by passion or negative binomial distribution.

Validity of the findings

The results were analyzed based on the available data, where maybe biased by the missing information. Also, the interviewees personal characteristic distribution should be discussed, such as age. gender.

The author may want to group some of the answers to avoid the low sample size issue.

The author claimed that the set of data analyzed is robust as it contains a large sample size, while as mentioned there are potential bias on the answers.

---

## Round 0.2 · accepted · Accept

As editor, I am satisfied the authors have addressed the reviewer's comments (including those they did not agree with), and addressed the issues I thought needed work. The manuscript is ready for publication (it's well-referenced) and I hope readers find it of interest.

Also of interest, is whether the Russian invasion of Ukraine and COVID to an extent, have led to a step change in attitudes (as raised in the Discussion). Publication at this juncture is worthwhile because it creates a record that any such changes can be measured against.